# Analysis and Verification of Finite Time Servo System Control with PSO Identification for Electric Servo System

**Zhihong Wu, Ruifeng Yang \*, Chenxia Guo, Shuangchao Ge and Xiaole Chen**

Automatic Test Equipment and System Engineering Research Center of Shanxi Province, North University of China, Taiyuan 030051, China; wuzhihong_0127@163.com (Z.W.); gchenxia@gmail.com (C.G.); geshch@nuc.edu.cn (S.G.); b1806013@st.nuc.edu.cn (X.C.)

**\*** Correspondence: yangruifeng@nuc.edu.cn; Tel.: +86-0351-3922619

**Abstract:** Electric servo system (ESS) is a servo mechanism in a control system of an aircraft, a ship, etc., which controls efficiency and directly affects the energy consumption and the dynamic characteristics of the system. However, the control performance of the ESS is affected by uncertainties such as friction, clearance, and component aging. In order to improve the control performance of the ESS, a control technology combining particle swarm optimization (PSO) and finite time servo system control (FTSSC) was introduced into ESS. In fact, it is difficult to know the uncertain physical parameters of the real ESS. In this paper, the genetic algorithm (GA) was introduced into PSO and the inertia weight was improved, which increased the parameter optimization precision and convergence speed. A new feedback controller is proposed to improve response speed and reduce errors by using FTSSC theory. The performance of the controller based on PSO identification algorithm was verified by co-simulation experiments based on Automatic Dynamic Analysis of Mechanical Systems (ADAMS) (MSC software, Los Angeles, CA, USA) and matrix laboratory (MATLAB)/Simulink (MathWorks, Natick, MA, USA). Meanwhile, the proposed strategy was validated on the servo test platform in the laboratory. Compared with the existing control strategy, the control error was reduced by 75% and the steady-state accuracy was increased by at least 50%.

**Keywords:** electric servo system; particle swarm optimization; finite time servo control; co-simulation

## 1. Introduction

Electric servo system (ESS) is a one kind of high precision position control system, for which control performance directly affects the stability and rapid response [1]. It is widely used in all-electric aircrafts and spacecrafts due to its high reliability and low energy consumption compared to hydraulic and pneumatic steering gears. Figure 1 shows the structure of an electric steering gear, which consists of a motor, a reduction gear, a screw, and a rocker arm. However, the difficulty of ESS is that its control performance is adversely affected by uncertainties such as friction, clearance, and part aging. Due to the complexity of the ESS, it is difficult to obtain accurate parameters directly from the actual system and the accuracy of system parameters is an important basis for accurate design and efficient application of the controller. Therefore, the premise for obtaining a high performance controller is to obtain accurate model parameters of ESS through optimization criteria. In this paper, the optimization criterion of ESS was to study the parameter identification strategy to find the optimal solution of the system model parameters in a dimensional space, so as to design an ESS controller with small control error, high steady-state accuracy, and low energy consumption. In order to solve the influence of uncertain parameters in the design of ESS controller, the optimization criteria of this study has made efforts from two aspects.

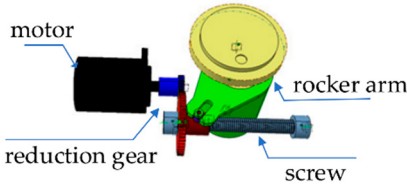

**Figure 1.** Internal schematic of the electric servo.

On the one hand, identification techniques were introduced to determine the actual physical parameters of the ESS. Since the mechanical structure of ESS is affected by uncertain factors such as friction and clearance, we cannot design the controller according to the theoretical parameters in practical applications. Therefore, the identification method based on the optimization algorithm has been widely used in ESS systems. For example, classic optimization algorithms have a large number of applications in system identification [2–4]. In [5], the classical recursive extended least squares (RELS) method was used to identify the model parameters of the ESS, and the experimental results showed that the improved least squares method can obtain accurate ESS model parameters. However, the least squares method must guarantee the derivation of the objective function and the unbiased estimation can be obtained only when the interference is white noise. Meanwhile, in view of the shortcomings and limitations of traditional system identification methods, intelligent optimization algorithms have been widely used in parameter optimization [6–10]. Wang developed an ESS model parameter identification strategy based on adaptive genetic algorithm (AGA). The experimental results showed that AGA can achieve better parameter optimization accuracy compared with other classical optimization algorithms. However, compared to other intelligent algorithms, this reference used a cumbersome encoding and decoding process based on binary coded AGA, which may lead to the complexity of the problem [11]. In addition, hybrid optimization algorithm is also applied to ESS because it combines the advantages of intelligent algorithms. In [12], Gao et al. studied a method for establishing an error model for industrial robots' ESS in order to obtain accurate model parameters. The ESS physical parameters of industrial robots were identified by a combination of PSO and BP neural network (BPNN) algorithms. Experimental results showed that the hybrid algorithm can obtain more accurate ESS parameters than PSO and GA.

On the other hand, the effects of uncertain factors are addressed by special control methods. Reference [13] studied the traditional proportional-integral-derivative (PID) control method and developed the parameter-adjustable nonlinear feedforward compensation method. However, when the ESS was in dynamic fast motion, the controller's transient accuracy and static tracking performance was reduced. In [14], a second-order aliasing surface sliding mode controller with switching gain was applied to the adjustment mechanism of the gain and position tracking control of the ESS. The experimental results showed that the optimization accuracy and response time of the controller were better than the sliding mode control. Reference [15] used discontinuous robust adaptive control methods to suppress system interference and achieve accurate tracking of the robot's ESS, but simple information fuzzy processing can reduce the control optimization accuracy and dynamic quality of the system. Inspired by the advantages of adaptive control and sliding mode control, an adaptive control method based on PID sliding mode was applied to ESS [16], the ESS experiment based on electronic throttling showed that the control optimization accuracy of this method was better than that of PID control [13] and H∞ control [17].

However, with the advocacy of the energy economy and the more accurate use of ESS, more effective control strategies have been put forward with strict requirements. The closed-loop systems under the above-mentioned controllers all have Lipschitz continuity and the fastest exponential form asymptotically converges, so these control analysis and synthesis methods are both infinitely stable control methods. Fortunately, from the perspective of system convergence time optimization, the controller based on finite-time convergence control has better control performance and is the time optimal control method [18–20]. In addition, finite-time theory has achieved results in many aspects [21], reference [22] studied a class of adaptive finite-time feedback devices based on

single-input/single-output (SISO). Experiments showed that the optimization accuracy of the state convergence rate method in a given time is higher than that of the set exponential method. In [23], a finite-time output feedback control method was applied to the position tracking system of the servo system. Experiments showed that the proposed optimization algorithm had faster tracking speed and higher tracking accuracy. In [24], a technique with time-varying delay based on finite-time stability theory was applied to ESS. The results showed that the improved optimization method was effective and feasible for theoretical application. As far as we know, it is a rare and valuable method to design an ESS controller by combining intelligent identification technology based on optimization algorithms with finite-time control theory. In [25], the method combining parameter identification and finite-time theory to optimize control has been successfully applied to ESS. Experiments show that the finite-time controller based on model parameter identification has better precision than the adaptive PID controller. Moreover, transforming system parameter identification into parameter optimization problem is an effective method to solve complex optimization problems by using optimized intelligent algorithms [26–28].

Inspired by the above literature research, in this paper, firstly, the physical parameters of ESS were identified by introducing particle swarm optimization (PSO) into genetic algorithm (GA) and improving inertia weight. Then, the control strategy of the finite time servo system was studied and the stability performance of the designed controller was proven. Finally, the PSO-based optimization identification technology was combined with the finite-time theory and the effectiveness of the method was verified by co-simulation and actual ESS test platform.

The remainder of this paper is organized as follows: Section 2 briefly illustrates the ESS mathematical model and describes some of the lemma knowledge of controller design. In Section 3, the recognition algorithm based on PSO and the FTSSC are given. The stability and convergence of ESS and control strategy are proven by co-simulation experiments based on ADAMS and MATLAB/Simulink. The experimental verification is shown in Section 4 by the experimental results carried out on the ESS test platform for several common situations in the flight application of the ESS. At last, Section 5 presents the conclusions.

## 2. Model of Electric Servo System

### 2.1. System Description

Electric servo system is a complex closed-loop system, which mainly includes controller, drive, servo motor, speed reduction mechanism, and feedback potentiometer. Figure 2 shows the structure of electric steering gear. The fundamental task and function of the electric steering gear is to enable the rudder axle to achieve aircraft attitude control according to a given speed and motion trajectory, which can ensure that the servo system can drive the load. When the desired rudder angle is given, it is compared with the actual rudder angle to generate a deviation signal, which is driven by the controller to drive the motor to rotate. The motor drives the rudder blade through the reducer to deflect in the direction required; when the actual rudder angle is equal to the desired rudder angle, the system reaches a new equilibrium state and the motor stops rotating to achieve angular position tracking. The controlled object is the rudder angle, the input is the desired rudder angle, and the output is the actual rudder angle.

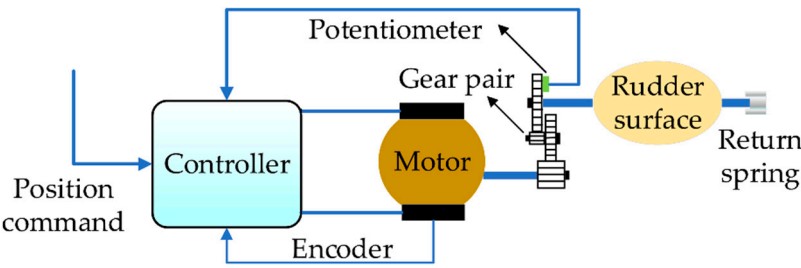

**Figure 2.** Block diagram of the electric servo system.

　　　The servo motor is the actuator of the ESS, and its schematic diagram is shown in Figure 3. The back EMF generated by the permanent magnet in the armature of the motor is as follows in Equation (1):

$$v_b = K_e w_m \tag{1}$$

where $w_m$ is the rotation speed, $K_e$ is the Back-emf constant.

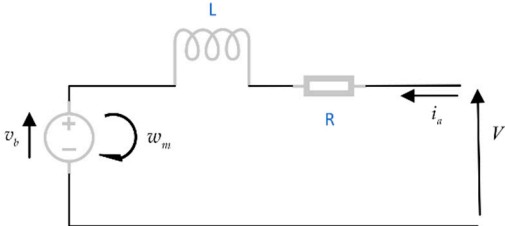

**Figure 3.** Schematic of the servo motor.

　　　As shown in Figure 3, the electric drive winding voltage balance equation is:

$$u = L\frac{di_a}{dt} + Ri_a + v_b \tag{2}$$

where $u$ is the control signal voltage, $i_a$ is the armature current, $R$ and $L$ are the resistance and inductance, respectively.

　　　The torque generated by the motor is proportional to the armature current $i_a$:

$$T_E = K_t i_a \tag{3}$$

where $K_t$ is the torque constant. Therefore, the torque equation of the motor shaft and the torque equation of the rudder shaft are Equations (4) and (5), respectively:

$$J_m \dot{w}_m = T_E - B_m w_m - T_m \tag{4}$$

$$J_t \dot{w} = T_1 - B_t w - T_{sp}(\theta) - T_f(w) \tag{5}$$

where $T_m$ and $T_1$ are the input and output torques, respectively. $J_m$, $B_m$ and $J_t$, $B_t$ are the moment inertia and viscous damping constant of the motor and rudder axle, respectively. $\theta$ and $w$ are the rotation angle and rotation speed of the servo system torque, respectively. $T_f$ is the friction torque, $T_{sp}$ is the return-spring torque. They are as follows:

$$T_f = F_c sgn(w) \tag{6}$$

$$T_{sp} = T_{LH} sgn(\theta - \theta_0) + k_s(\theta - \theta_0), \theta_{\min} \le \theta \le \theta_{\max} \tag{7}$$

where $F_c$ is the friction coefficient, $T_{LH}$ and $k_s$ are the offset and gain, respectively. $\theta_0$ is the default initial deflection angle of the ESS. $Sgn(.)$ is the sign function defined as:

$$sgn(s) = \begin{cases} 1 & for\ s > 0 \\ 0 & for\ s = 0 \\ -1 & for\ s < 0 \end{cases} . \tag{8}$$

　　　In practical applications, the physical parameters $L$, $R$, $k_s$, $T_{LH}$, $F_c$, $J$, $B$, $K_t$, $K_e$, and $n$ are not fully known and there is a large degree of uncertainty, especially, such as $k_s$, $T_{LH}$, $F_c$. Therefore, for cases where system parameters are susceptible to these complex factors, mathematical model design control methods based on system parameter identification are often used.

Since the armature inductance value $L$ is small, the armature current dynamics $i_a$ in Equation (2) can be neglected, so it can be considered as $u = v_b = K_e w_m$. The dynamic equation of the main body of the ESS can be equated with the following simple state space form of the second-order system in the controller design:

$$
\begin{cases}
\dot{\theta}_m = w_m \\
\dot{\theta} = w \\
\dot{w} = -a_1(\theta - \theta_0) - a_2 w + bK_e w_m - c_1 sgn(\theta - \theta_0) - c_2 sgn(w)
\end{cases}
\tag{9}
$$

where $a_1 = \frac{k_s}{J}$, $a_2 = \frac{BR + n^2 K_t K_e}{JR}$, $b = \frac{nK_t}{JR}$, $c_1 = \frac{T_{LH}}{J}$, $c_2 = \frac{F_c}{J}$, $J = n^2 J_m + J_t$, $B = n^2 B_m + B_t$.

## 2.2. Fundamental Lemma

Since the finite time control system does not satisfy the Lipschitz continuity, the control design tool established under the Lipschitz continuity condition is no longer suitable for the analysis and design of the finite time control method. Therefore, it is necessary to introduce the relevant theory of finite time control before the control design:

**Lemma 1** ([20]). Assume the following non-Lipschitz continuous system

$$
\dot{x} = f(x), \ f(0) = 0, \ x \in \mathbb{R}^n
\tag{10}
$$

suppose there exists a continuous function $V: U \to \mathbb{R}$ such that the following conditions hold:

(1)  $V(x)$ is a positive definite and continuous function on the domain $U$.
(2)  There exist real numbers $c > 0, \alpha \in (0, 1)$ and an open neighborhood $U_0 \subset U$ containing the origin, so that the following conditions are true

$$
\dot{V} + cV^\alpha(x) \le 0, \ x \in U_0 \backslash \{0\}.
\tag{11}
$$

For system (10), it is finite-time stable if it is Lyapunov stable in neighborhood $U \subset U_0$ and its state can converge to equilibrium point $x = 0$ within a finite time. A more precise description is that if there is a function $T(x_0)$: $U\backslash\{0\} \to (0, \infty)$, so that for $\forall x_0 \in U \subset U_0$, the system is decoded as $x(t, x_0)$; For $x(t, x_0) \in U\backslash\{0\}$ and $lim_{t \to \bar{T}(x_0)} x(t, x_0) = 0$ when $t \in [0, T(x_0)]$, for $t > T(x_0)$, there is $x(t, x_0) = 0$, then the system is stable for local finite time. On the basis of this, if $U = U_0 = \mathbb{R}^n$, the system is global finite-time stable.

**Lemma 2** ([20]). For any real numbers $x_i$, $i = 1, \dots, n$, $y$ and $0 < q \le 1$, the following inequalities hold:

$$
(|x_1| + \dots + |x_n|)^q \le |x_1|^q + \dots + |x_n|^q
\tag{12}
$$

$$
\left| x^q - y^q \right| \le 2^{1-q} \left| x - y \right|^q
\tag{13}
$$

when $q = q_1/q_2 \le 1$, $q_1$ and $q_2$ are odd integers.

**Lemma 3** ([20]). Let $c, d \in R^+$ and $\gamma(x, y) > 0$ be a real-valued function. Then,

$$
|x|^c |y|^d \le \frac{c\gamma(x,y)|x|^{c+d}}{c+d} + \frac{d\gamma^{-c/d}(x,y)|y|^{c+d}}{c+d}.
\tag{14}
$$

## 3. Control Strategy Based on PSO Identification

### 3.1. Parameter Identification Based on PSO

　　PSO algorithm is an evolutionary intelligence algorithm, which originates from birds' undirected clustering flight model. It is proposed on the basis of the similarity between the behavior characteristics of a bird swarm and the solution of optimization problems [26]. In the algorithm, it is assumed that a particle population is distributed in a $D$ dimensional search space according to a certain rule and each particle flies at a certain initial velocity from the starting position. Select the appropriate number of particles $m$ as the group $X$. Each particle is a $D$-dimensional vector $X_i = [x_{i1}, x_{i2}, \ldots, x_{iD}]^T$, which represents a solution to the optimization object. $V_i = [v_{i1}, v_{i2}, \ldots, v_{iD}]^T$ represents the flight speed of the $i$th particle. $P_i = [P_{i1}, P_{i2}, \ldots, P_{iD}]^T$ and $P_g = [P_{g1}, P_{g2}, \ldots, P_{gD}]^T$ represent individual optimal and global optimal, respectively. The individual updates its individual optimal position at any time according to its own flight history and can obtain the global optimal information of the group at any time according to the group information sharing. In each update iteration, the particle tracks the above two types of optimal information to change its flight trajectory. The speed and position update formula is as follows:

$$\begin{cases} V_{id}^{k+1} = wV_{id}^k + c_1 r_1 (P_{id}^k - X_{id}^k) + c_2 r_2 (P_{gd}^k - X_{gd}^k) \\ X_{id}^{k+1} = X_{id}^k + V_{id}^{k+1} \end{cases} \tag{15}$$

where $d = 1, 2, \ldots, D$. $i = 1, 2, \ldots, m$. $k$ represents iteration steps, $k = 0, 1, \ldots, k_m$. $w$ is an inertia weight. $c_1, c_2$ are non-negative constants named the acceleration factors, $r_1, r_2$ are random numbers distributing on [0,1].

　　The identification diagram is shown in Figure 4. By considering the combination of identification accuracy and optimization speed, choose $m = 40$, $k_m = 120$, and $X_i = [L, R, k_s, T_{LH}, F_c, J, B, nK_t, nK_e]^T$. The fitness function is selected by the ISE criterion. The fitness function is defined as follows:

$$ISE = \int (\theta_a - \theta_i)^2 dt \tag{16}$$

where $\theta_a$ and $\theta_i$ represent the real and identified angle of the ESS, respectively. The smaller the fitness value, the closer the model consisting of the identification parameters is to the actual system model.

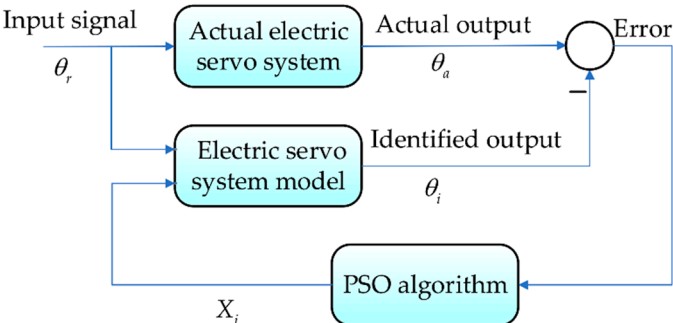

**Figure 4.** Identification schematic.

### 3.2. Improvements on PSO

　　According to the above description, it can be known that the particles in the PSO only share information through the current search to the best point, which will cause the particles (the identified system parameters) to fall into the local optimum. When $P_g$ maintains the same value for five iterations in the iterative process but the criterion is not within the allowable error, GA is used to improve the search range and search ability of PSO, thus avoiding particles that are "premature". In order to avoid the misunderstanding of the proposed solution (optimization process), which is only for specific laboratory cases, we explain the value: if this value is too small, then the optimization speed cannot be

guaranteed and there will be a phenomenon whereby PSO is not "premature" but frequently uses GA for extensive search; if this value is too large, then the optimization accuracy cannot be guaranteed and there will be multiple invalid cycles of PSO, but GA cannot help to enhance the search ability.

### 3.2.1. Introducing GA into PSO

The principle of GA is like natural evolution. According to the rules of the survival of the fittest, only the genes of individuals adapting to the environment can survive and pass on to the next generation, and the individual gradually develops into an excellent individual [29]. The procedure is as follows:

(1) Crossover. It is assumed that the crossover probability is $P_c$ in the entire population and the crossover operations are performed between individuals when the crossover probability is greater than the set value. The offspring $X_i^{k+1}$ and $X_j^{k+1}$ of randomly chosen parents $X_i^k$ and $X_j^k$ are:

$$if\ rand1 > P_c\ then\ X_i^{k+1} = X_i^k + rand \times \left(X_j^k - X_i^k\right) X_j^{k+1} = X_j^k + rand \times \left(X_i^k - X_j^k\right);$$
$$else\ X_i^{k+1} = X_i^k\ X_j^{k+1} = X_j^k; \tag{17}$$

where *rand* and *rand*1 represent random numbers that obey the uniform distribution.

(2) Mutation. Assume that the crossover probability $P_m$ is greater than the set value. The offspring population is generated according to Equation (18):

$$if\ rand2 > P_m\ then$$
$$if\ r < 0.5\ then\ X_i^{k+1} = X_i^k + \lambda(U_k - L_k)\ else\ X_i^{k+1} = X_i^k - \lambda(U_k - L_k); \tag{18}$$
$$else\ X_i^{k+1} = X_i^k;$$

where *rand*2, *r*, and $\lambda$ represent random numbers that obey the uniform distribution, the element $X_i^k \in [L_k, U_k]$.

(3) Replacement. Calculate the individual fitness values of the offspring after the crossover and mutation, and the elite retention strategy according to Equation (19):

$$if\ ISE_{X_i^k} < ISE_{X_i^{k+1}}\ then$$
$$P_g = \left\{ P_i \middle| ISE_{X_i^k} = min,\ i \in \{1, 2, \ldots, m\} \right\} \tag{19}$$
$$else\ P_g = \left\{ P_i \middle| ISE_{X_i^{k+1}} = min,\ i \in \{1, 2, \ldots, m\} \right\}$$

### 3.2.2. Improve Inertia Weight

The schematic diagram of the improved PSO is shown in Figure 5. The inertia weight and acceleration factors play a key role in the global and local performance of the PSO. In the early stage of evolution, in order to speed up the search ability, a large inertia weight $w$ is selected to expand the optimization space. When the algorithm evolves to the later stage, the local search is strengthened by reducing the inertia weight $w$ to speed up the convergence. According to the above description, gradually reducing the acceleration coefficient $c_1$ and gradually increasing $c_2$ meets the early and late requirements of the algorithm optimization [30]. Therefore, choose inertial weight and acceleration coefficient as:

$$\begin{cases} w = w_{\min} + (w_{\max} - w_{\min})e^{-20(k/k_m)^6} \\ c_1 = 0.5w^2 + w + 1,\ \ c_2 = 2.5 - c_1 \end{cases} \tag{20}$$

with $w_{\max} = 0.9$ and $w_{\min} = 0.4$.

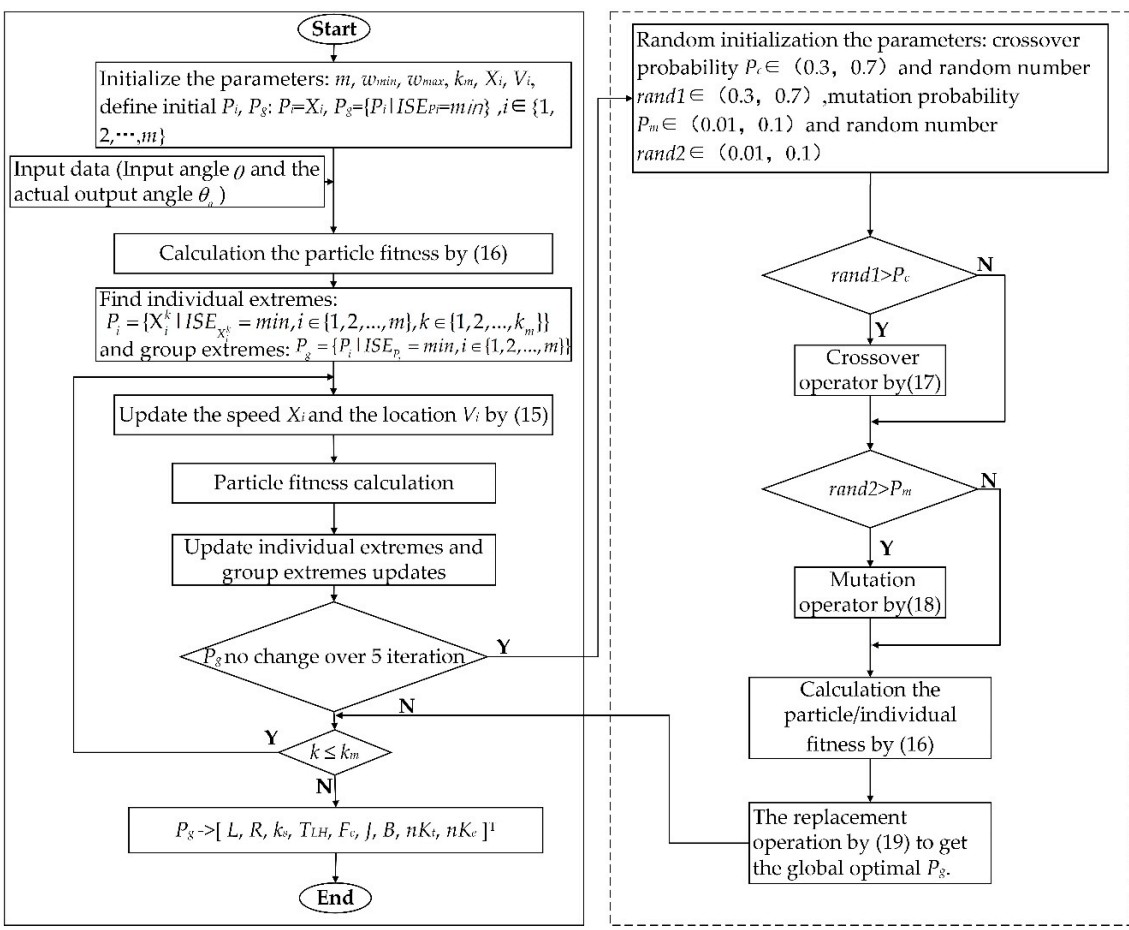

**Figure 5.** Flowchart of the improved particle swarm algorithm.

### 3.3. Controller Based on Finite Time

For an ESS, a good motor drive system must be able to control the steering gear with high response accuracy and response speed, and be able to resist some external disturbances. By identifying the results, the ESS servo controller can be designed using the finite time theory of the servo system based identification model. Therefore, for the model of the identified system (9), the following FTSSC was designed:

$$u = b^{-1}\left\{-a_1 x_2 - a_2 x_3 + k_3 \left[x_3^{\frac{1}{2q-1}} + k_2^{\frac{1}{2q-1}}\left(x_2^{\frac{1}{q}} + k_1^{\frac{1}{q}} x_1\right)\right]^{3q-2}\right\} \tag{21}$$

where $x_1 = \int (\theta_r - \theta)dt$, $x_2 = \theta_r - \theta$, $x_3 = \dot{x}_2 = \dot{\theta}_r - w$, $\theta_r$ is the desired reference input; the adjustable constants $k_1$, $k_2$, and $k_3$ are positive; and $q = q_1/q_2 \in (2/3, 1)$, $q_1 > 0$, and $q_2 > 0$ are odd integers, which satisfy the following conditions:

$$m_1 = k_1 - \frac{l(3-2q)k_2^{1/(2q-1)}}{1+q}\left[\gamma_5 2^{2-2q}qk_1^{1+1/q} + \gamma_7 M_3\right] - 2^{1-q}\left[\gamma_1 + \gamma_2(2-q)k_1^{1+1/q}\right]/(1+q) > 0 \tag{22}$$

$$\begin{aligned}m_2 &= k_2 - 2^{1-q}(2-q)k_1^{1/q} - \gamma_1^{-1/q}2^{1-q}q/(1+q) - (2-q)\left[\gamma_2^{-q}2^{1-q}k_1^{1+1/q} + \gamma_3 2^{2-2q}\right]/(1+q) \\ &\quad - \frac{l(3-2q)k_2^{1/(2q-1)}}{1+q}\left[\gamma_4 2^{2-2q}qk_1^{1/q} + \gamma_6 M_2\right] > 0\end{aligned} \tag{23}$$

$$m_3 = lk_3 - \gamma_3^{-(2-q)/(2q-1)} 2^{2-2q}(2q-1)/(1+q) - \frac{l(\gamma_4^{-q}+\gamma_5^{-q}k_1)2^{2-2q}(3-2q)k_1^{1/q}k_2^{1/(2q-1)}}{1+q}$$
$$-\frac{l(3-2q)k_2^{1/(2q-1)}}{q}\left[M_1 + \frac{\gamma_6^{-q}M_2}{1+q} + \frac{\gamma_7^{-q}M_3}{1+q}\right] > 0 \tag{24}$$

where $l, \gamma_i (i = 1, \ldots, 7)$ are positive adjustable constants.

According to the definition of servo controller (21) and state variable $x = [x_1, x_2, x_3]^T$ designed in this paper, system (9) is described as:

$$\begin{cases} \dot{x}_1 = x_2 \\ \dot{x}_2 = x_3 \\ \dot{x}_3 = -k_3\left[x_3^{1/(2q-1)} + k_2^{1/(2q-1)}\left(x_2^{1/q} + k_1^{1/q}x_1\right)\right]^{3q-2} \end{cases} . \tag{25}$$

The time derivative $\dot{V}_1(x_1) = \frac{1}{2}x_1^2$ of the Lyapunov function considered for the nonlinear state space model (25) was chosen to satisfy the following formula:

$$\dot{V}_1(x_1) \le |x_1(x_2 - x_{2d})| - k_1 x_1^{1+q} \tag{26}$$

where $x_{2d}$ is the virtual control law and is constructed as $x_{2d} = -k_1 x_1^q$. For the subsystem $(x_1, x_2)$ of Equation (25), a Lyapunov function of the power integrator was chosen as follows:

$$V_2(x_1, x_2) = V_1(x_1) + \int_{x_{2d}}^{x_2} (s^{\frac{1}{q}} - x_{2d}^{\frac{1}{q}})^{2-q} ds. \tag{27}$$

Therefore, the time derivative of $V_2(x_1, x_2)$ is obtained as

$$\dot{V}_2 \le |x_1(x_2 - x_{2d})| - k_1 x_1^{1+q} + \varsigma_2^{2-q}x_3 + (2-q)k_1^{\frac{1}{q}}x_2\int_{x_{2d}}^{x_2}(s^{\frac{1}{q}} - x_{2d}^{\frac{1}{q}})^{1-q}ds$$
$$\le |x_1(x_2 - x_{2d})| - k_1 x_1^{1+q} + \varsigma_2^{2-q}(x_3 - x_{3d}) + \varsigma_2^{2-q}x_{3d} + (2-q)k_1^{\frac{1}{q}}|x_2||x_2 - x_{2d}|\varsigma_2^{1-q} \tag{28}$$

where $\varsigma_2 = x_2^{1/q} - x_{2d}^{1/q}, \partial(-x_{2d}^{1/q})/\partial x_1 = k_1^{1/q}$ are utilized. The Lemmas 2 and 3 are applied to the first half and the second half of Inequality (28), and the final results are as follows:

$$|x_1(x_2 - x_{2d})| \le 2^{1-q}|x_1||\varsigma_2^q| \le 2^{1-q}(\gamma_1 x_1^{1-q} + \gamma_1^{-1/q}q\varsigma_2^{1+q})/(1+q) \tag{29}$$

$$|x_2||x_2 - x_{2d}| \le 2^{1-q}|\varsigma_2 + x_{2d}^{1/q}|^q|\varsigma_2^q| \le 2^{1-q}\varsigma_2^{2q} + 2^{1-q}k_1|x_1^q||\varsigma_2^q|. \tag{30}$$

Then, substituting (29) and (30) into (28) yields

$$\dot{V}_2 \le 2^{1-q}(\frac{\gamma_1 x_1^{1+q}}{1+q} + \frac{\gamma_1^{-1/q}q\varsigma_2^{1+q}}{1+q}) - k_1 x_1^{1+q} + \varsigma_2^{2-q}(x_3 - x_{3d}) + 2^{1-q}(2-q)k_1^{1/q}\varsigma_2^{1+q}$$
$$+\varsigma_2^{2-q}x_{3d} + 2^{1-q}(2-q)k_1^{1+1/q}|x_1^q||\varsigma_2| \tag{31}$$

Moreover, according to Lemma 3

$$|x_1|^q|\varsigma_2| \le \frac{\gamma_2 q x_1^{1+q}}{1+q} + \frac{\gamma_2^{-q}\varsigma_2^{1+q}}{1+q}. \tag{32}$$

Additionally, let $x_{3d} = -k_2\varsigma_2^{2q-1}$, Inequality (31) becomes

$$\dot{V}_2 \le \left[-k_2 + 2^{1-q}\left[\frac{\gamma_1^{-1/q}q}{1+q} + (2-q)k_1^{1/q}(1 + \frac{\gamma_2^{-q}k_1}{1+q})\right]\right]\varsigma_2^{1+q}$$
$$+\left[-k_1 + \frac{2^{1-q}}{1+q}(\gamma_1 + \gamma_2 q(2-q)k_1^{1+1/q})\right]x_1^{1+q} + |\varsigma_2|^{2-q}|x_3 - x_{3d}| \tag{33}$$

The constructed Lyapunov function is as follows

$$V_3(x) = V_2(x_1, x_2) + l \int_{x_{3d}}^{x_3} \left( s^{\frac{1}{2q-1}} - x_{3d}^{\frac{1}{2q-1}} \right)^{3-2q} ds. \tag{34}$$

Noting that $\partial(-x_{3d}^{\frac{1}{2q-1}})/\partial x_1 = k_1^{\frac{1}{q}} k_2^{\frac{1}{2q-1}}, \partial(-x_{3d}^{\frac{1}{2q-1}})/\partial x_2 = \frac{1}{q} k_2^{\frac{1}{2q-1}} x_2^{\frac{1}{q}-1}$, the time derivative of $V_3(x_1, x_2, x_3)$ is

$$\dot{V}_3(x) \le \dot{V}_2 + l\varsigma_3^{3-2q} \dot{x}_3 + l(3-2q) k_2^{\frac{1}{2q-1}} |x_3 - x_{3d}|\varsigma_3^{2-2q} \times \left[ k_1^{\frac{1}{q}} |x_2| + \frac{1}{q} \left| x_2^{\frac{1}{q}-1} \right| |x_3| \right] \tag{35}$$

where $\varsigma_3 = x_3^{1/(2q-1)} - x_{3d}^{1/(2q-1)}$. By applying Lemmas 2 and 3, the following inequalities are established:

$$|\varsigma_2|^{2-q}|x_3 - x_{3d}| \le 2^{2-2q}|\varsigma_2|^{2-q}|\varsigma_3|^{2q-1} \le \frac{2^{2-2q}}{1+q} \left[ \gamma_3(2-q)\varsigma_2^{1+q} + \gamma_3^{-\frac{2-q}{2q-1}}(2q-1)\varsigma_3^{1+q} \right] \tag{36}$$

$$|x_2||x_3 - x_{3d}|\varsigma_3^{2-2q} \le 2^{2-2q}(|\varsigma_2|^q|\varsigma_3| + k_1|x_1|^q|\varsigma_3|) \le \frac{2^{2-2q}}{1+q} \left[ \gamma_5 q k_1 x_1^{1+q} + \gamma_4 q \varsigma_2^{1+q} + (\gamma_4^{-q} + \gamma_5^{-q} k_1)\varsigma_3^{1+q} \right] \tag{37}$$

$$\left| x_2^{\frac{1}{q}-1} \right| |x_3| \le \left[ \frac{1-q}{q} + k_2 + \frac{(2q-1)k_1^{\frac{1}{q}-1} k_2}{q} \right] |\varsigma_2|^q + \frac{(2q-1)(1+k_1^{\frac{1}{q}-1})}{q}|\varsigma_3|^q + \frac{(1-q)k_1^{\frac{1}{q}-1}(1+k_2)}{q}|x_1|^q \tag{38}$$

Furthermore

$$\begin{aligned} \left| x_2^{1/q-1} \right| |x_3||x_3 - x_{3d}|\varsigma_3^{2-2q} &\le M_1\varsigma_3^{1+q} + M_2|\varsigma_2|^q|\varsigma_3| + M_3|x_1|^q|\varsigma_3| \\ &\le M_1\varsigma_3^{1+q} + M_2 \left[ \frac{\gamma_6 q}{1+q}\varsigma_2^{1+q} + \frac{\gamma_6^{-q}}{1+q}\varsigma_3^{1+q} \right] + M_3 \left[ \frac{\gamma_7 q}{1+q}x_1^{1+q} + \frac{\gamma_7^{-q}}{1+q}\varsigma_3^{1+q} \right] \end{aligned} \tag{39}$$

After that, (33), (36), (37), and (39) were substituted into (35), and at the same time, the definitions of $\varsigma_3, x_{3d}, \varsigma_2, x_{2d}$ and $m_1, m_2, m_3$ of the systems (25) and (22)–(24) were considered, respectively. $\dot{V}_3$ satisfies the following formulas:

$$\dot{V}_3(x) \le -m_1 x_1^{1+q} - m_2 \varsigma_2^{1+q} - m_3 \varsigma_3^{1+q}. \tag{40}$$

Meanwhile, by Lemma 2 $V_3(x_1, x_2, x_3)$ satisfies

$$V_3(x) \le \frac{1}{2}x_1^2 + |x_2 - x_{2d}|\varsigma_2^{2-q} + |x_3 - x_{3d}|\varsigma_3^{3-2q} \le 2x_1^2 + 2\varsigma_2^2 + 2\varsigma_3^2. \tag{41}$$

Then, by Lemma 2, the following inequality holds.

$$\dot{V}_3(x) \le -2^{-\frac{1+q}{2}}\lambda\left(2x_1^2 + 2\varsigma_2^2 + 2\varsigma_3^2\right)^{\frac{1+q}{2}} \le -2^{-\frac{1+q}{2}}\lambda V_3(x_1, x_2, x_3)^{\frac{1+q}{2}} \tag{42}$$

where $\lambda = \min\{m_1, m_2, m_3\}$, through the above description, and finally through Lemma 1, from (34) and (42) it was concluded that the system (25) is stable with $c = 2^{-\frac{1+q}{2}}\lambda, \alpha = \frac{1+q}{2} \in (0,1)$ for a finite time at the equilibrium point $x = 0$. In short, the proposed control strategy (21) ensures that the system (9) has a desired angle of higher static performance $x_1 = \int (\theta_r - \theta)dt = 0, x_2 = \theta_r - \theta = 0$ for a limited time.

## 3.4. Co-Simulation

In order to prove the performance of the proposed FTSSC, co-simulation experiments based on ADAMS and MATLAB/Simulink were implemented. As shown in Figure 6a, the mechanical dynamics analysis software ADAMS was used to simulate the dynamics of the steering gear, and

the motion parameters of the dynamic model were provided to Simulink. Simulink receives these external parameters and outputs the controller model to the ADAMS. External excitation (control force or torque) is required to control the movement of the mechanism components.

As shown in Figure 6b, the ADAMS–Simulink interface module generated by ADAMS/Control motor model and FTSSC model were combined to establish the servo control system. The pink module in Figure 6b is a nonlinear subsystem constructed by ADAMS. The model function is to realize the rudder angle position tracking. The actual rudder angular displacement of the servo motor output is used as feedback and the model motion in the ADAMS is controlled by the torque output of the motor model. For example, when a servo system input signal $\theta$ is given, firstly, the controller outputs a voltage signal $U_m$ based on the FTSSC strategy proposed in this paper. Then, the voltage $U_m$ output by the controller and the steering feedback rotational speed $w_m$ are used as the input to the motor, so that the motor outputs the torque $T_m$. Finally, the torque drives the mechanical model structure based on the ADAMS, thereby outputting the deflection angle $\theta_a$ and the rotational speed $w_m$.

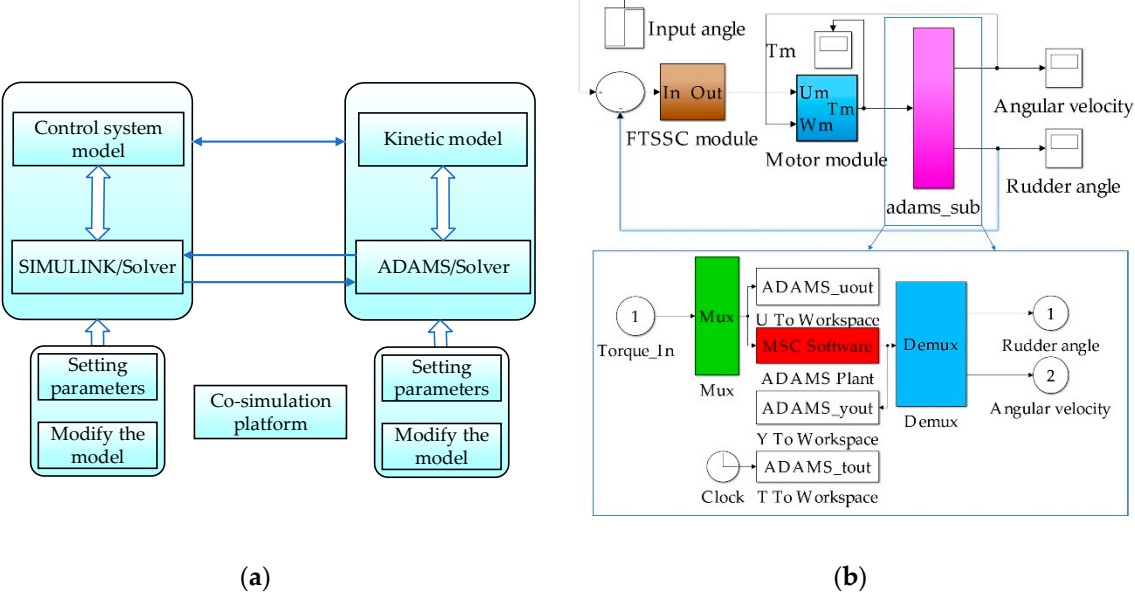

(**a**)　　　　　　　　　　　　　　　　(**b**)

**Figure 6.** Co-simulation based on ADAMS and Simulink: (**a**) the scheme of co-simulation; (**b**) the scheme of MALTAB/Simulink.

The principle of the simulation experiment platform is introduced above. In order to verify the superiority of our proposed strategy, we compared the controller proposed in Section 3.3 with the existing strategy through the simulation under this environment. In [16], the robust adaptive PID sliding mode control (RASMC) strategy was used to verify the transient performance of the system and the robust performance, and the experimental results showed that the control performance of the proposed RASM controller was much better than the PID controller in [13] and the H∞ controller in [17]. Therefore, we only compared the FTSSC strategy proposed in this paper with the RASM control strategy in [16].

In the simulation, the system model was the same as in [16] and Table 1 lists the system parameter values in the co-simulation modeling. The parameters of the controller (21) are set as follows:

$$q = 96/97, k_1 = 0.25, k_2 = 38, k_3 = 400, l = 0.00019,$$
$$\gamma_1 = 0.03, \gamma_2 = 1, \gamma_3 = 20, \gamma_4 = 1, \gamma_5 = 15, \gamma_6 = 10, \gamma_7 = 54 \tag{43}$$

consequently, $m_1 = 0.0076$, $m_2 = 5.7595$, $m_3 = 0.00035$. Figures 7 and 8 show the response curve of the control command of the ESS, the controller (21), and the RASMC, respectively.

**Table 1.** Physical parameters of Electric servo system (ESS).

| Parameters | Value | Unit |
|:---:|:---:|:---:|
| $J$ | 0.0031 | kgm$^2$ |
| $B$ | 0.0098 | Nm s/rad |
| $K_t$ | 0.0175 | Nm/A |
| $K_e$ | 0.0295 | V s/rad |
| $k_s$ | 0.0877 | Nm/rad |
| $T_{LH}$ | 0.46 | Nm |
| $F_c$ | 0.34 | Nm |
| $R$ | 1.55 | $\Omega$ |
| $L$ | 1.6 | mH |
| $\theta_0$ | 15 | $^\circ$ |
| $n$ | 22.26 | / |

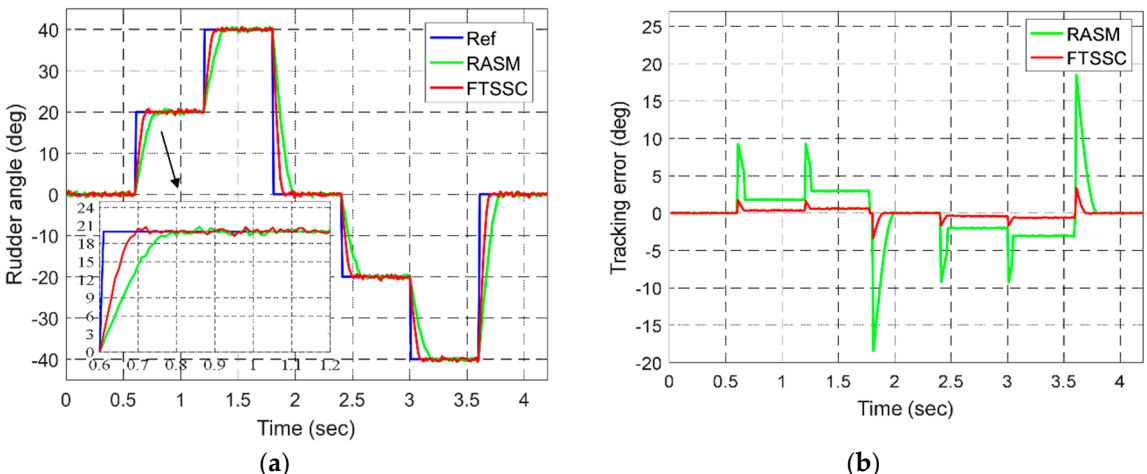

**Figure 7.** The input signal is a step signal: (**a**) the response curves; (**b**) tracking error.

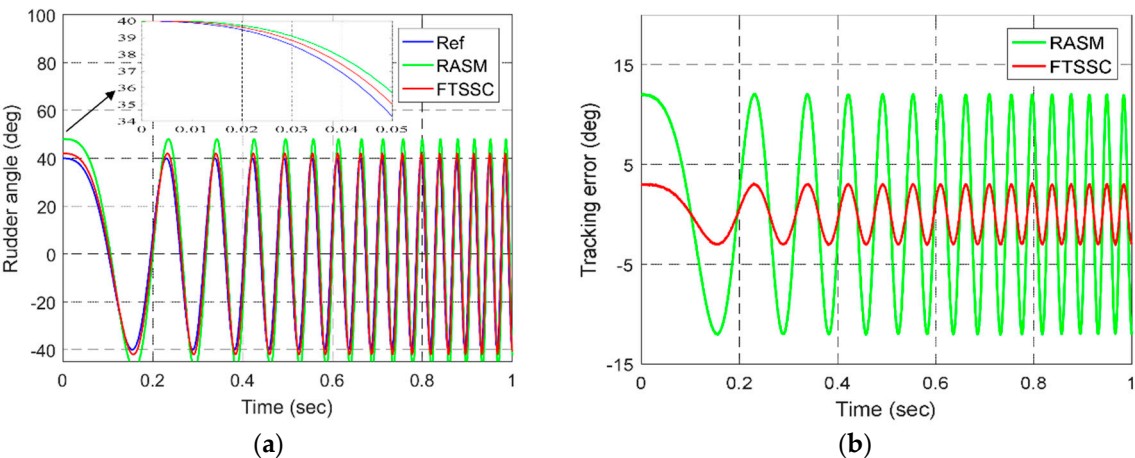

**Figure 8.** The input signal is a chirp signal: (**a**) the response curves; (**b**) tracking error.

In summary, Figures 7 and 8 show two distinct advantages of the PSO-based FTSSC strategy compared to the RASM controller. One is that the ESS has a fast response speed (response time is less than 100 ms), which indicates that the ESS tracking speed is fast. The other is that the controller tracking error based on FTSSC is smaller than the RASM control method, which indicates that the ESS tracking accuracy is very high.

## 4. Experimental Application and Results Analysis

This section verifies the effectiveness of applying the proposed strategy to the ESS. The experiment was performed on the test platform of the ESS shown in Figure 9. The experiment was divided into two parts—one was to carry out the system parameter identification experiment of ESS and the other part was to apply the identified parameters to the FTSSC controller designed in this paper.

### 4.1. Parameter Identification Experiment

The experimental flow of parameter identification is as follows: Firstly, the test equipment sends the desired angle signal and collects the feedback signal of ESS. The feedback signal is the real world output angle $\theta_a$ of ESS. Then, using the algorithm programming in MATLAB environment, the control signal data (expected angle) are input into the preset identification model to get the identification angle $\theta_i$. According to the definition of the fitness value of Equation (16), the ESS parameters are continuously optimized by the improved PSO proposed in this paper to make it close to the physical parameters of real-world real systems. Finally, the parameter identification performance of four different algorithms are compared. Therefore, as in the simulation experiment, the parameter identification optimization criterion of the actual system in this paper is also carried out through the definition of Equation (16).

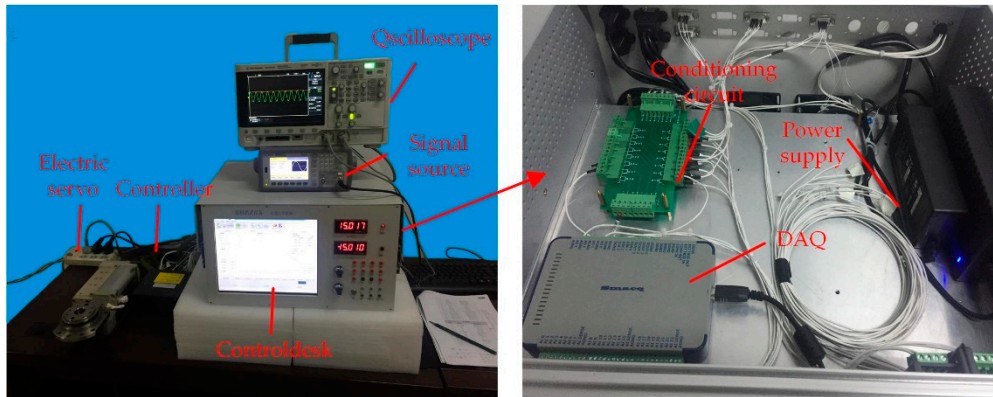

**Figure 9.** Electric servo system (ESS) hardware-in-loop test platform.

In order to verify the effectiveness of the proposed PSO, the other three optimization algorithms GA, PSO, and IPSO1 were compared with the improved PSO. Among the four algorithms, IPSO1 is a PSO algorithm without GA but with improved inertia weight, and IPSO2 is the proposed algorithm in this paper. For fair comparison, the basic parameters of all algorithms were set according to Section 3 and these four algorithms were compared with the fitness values defined in (16). In addition, the parameter error (PE) and the average parameter error (APE) were employed in this study to evaluate the parameter identification accuracy of system. The PE and APE were calculated as follows:

$$PE = \left| \frac{X_i - \overset{\wedge}{X}_i}{X_i} \right| \times 100\%, \ \ i = 1, 2, \dots, D \tag{44}$$

$$APE = \frac{1}{D} \sum_{k=1}^{D} \left| \frac{X_i - \overset{\wedge}{X}_i}{X_i} \right| \times 100\% \tag{45}$$

where $X_i$ is the parameter in actual system, $\overset{\wedge}{X}_i$ is the parameter in identified system, $D$ is the dimension of $X_i$.

Figure 10 shows the identification process of different algorithms. In order to avoid being misled by Figure 10, this paper only compared the optimization results of each algorithm, so that they can

avoid the irrelevant relationship caused by random data (at initialization). The results show that the IPSO2 has the advantages of higher convergence accuracy (smaller fitness value) and fewer iteration steps compared with the other three algorithms. Meanwhile, it can be seen from Table 2 that the maximum PE of IPSO2 is only 5%, far lower than the 18% of GA and 25% of PSO, which indicates that the improvement strategy, such as the introducing GA into PSO and improved inertia weight, can effectively improve the convergence accuracy and optimization speed of the algorithm.

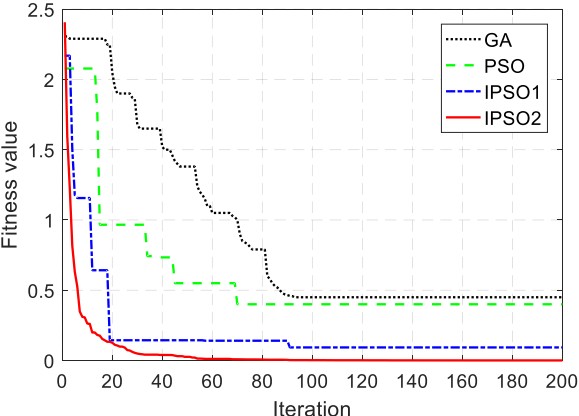

**Figure 10.** Identification process of different algorithms.

**Table 2.** ESS physical parameter identification results.

| Parameters | Theoretical Value | Mean Value of the Identification Parameter (30 Trials) | | | | | | | |
|---|---|---|---|---|---|---|---|---|---|
| | | GA | | PSO | | IPSO1 | | IPSO2 | |
| | | $\hat{X_i}$ | PE | $\hat{X_i}$ | PE | $\hat{X_i}$ | PE | $\hat{X_i}$ | PE |
| $L$ | 0.005 | 0.0041 | 0.18 | 0.0045 | 0.1 | 0.0047 | 0.06 | 0.0049 | 0.02 |
| $R$ | 1.5 | 1.53 | 0.02 | 1.55 | 0.033 | 1.52 | 0.013 | 1.508 | 0.005 |
| $k_s$ | 0.1 | 0.080 | 0.2 | 0.075 | 0.25 | 0.088 | 0.12 | 0.098 | 0.02 |
| $T_{LH}$ | 0.1 | 0.095 | 0.05 | 0.092 | 0.08 | 0.095 | 0.05 | 0.098 | 0.02 |
| $F_c$ | 0.01 | 0.008 | 0.2 | 0.008 | 0.2 | 0.009 | 0.1 | 0.0098 | 0.02 |
| $J$ | 0.004 | 0.0032 | 0.2 | 0.0035 | 0.125 | 0.0037 | 0.075 | 0.0038 | 0.05 |
| $B$ | 0.8 | 0.769 | 0.039 | 0.751 | 0.061 | 0.772 | 0.035 | 0.783 | 0.021 |
| $K_t$ | 0.93 | 0.89 | 0.043 | 0.902 | 0.03 | 0.925 | 0.005 | 0.934 | 0.004 |
| $K_e$ | 0.005 | 0.0044 | 0.12 | 0.0042 | 0.16 | 0.0045 | 0.1 | 0.0049 | 0.02 |
| **APE** | | 0.117 | | 0.115 | | 0.062 | | 0.02 | |

*4.2. Controller Verification*

The physical parameters of ESS were obtained by Section 4.1. In order to verify the transient response speed and steady-state accuracy of the proposed controller in practical application. We compared this with the existing feedforward PID control [13] and H∞ control law [17] as follows:

$$u_{PID} = -5x_2 - 1.2x_1 - 0.1x_3 + \tau_{fa0} + \tau_{spa0} \tag{46}$$

$$u_{H\infty} = 0.15\ddot{\theta}_r + 0.5\dot{\theta} - 14x_2 - 0.6x_3 \tag{47}$$

where $\tau_{fa} = \frac{F_c sgn(w)R}{nK_t}, \tau_{spa} = \frac{[T_{LH}sgn(\theta-\theta_0)+k_s(\theta-\theta_0)]R}{nK_t}$.

Through the above identification experiment of ESS, since the actual physical parameters are different from the co-simulation model, the parameters of the controller (21) were set as follows:

$$q = 96/97, k_1 = 0.1, k_2 = 16, k_3 = 233, l = 0.00015,$$
$$\gamma_1 = 0.14, \gamma_2 = 1, \gamma_3 = 19, \gamma_4 = 1, \gamma_5 = 15, \gamma_6 = 10, \gamma_7 = 10 \tag{48}$$
$$m_1 = 0.0185, m_2 = 1.6428, m_3 = 0.001$$

The experiment applied three different reference signals to the ESS on the actual hardware test equipment of the ESS. The experimental conditions are as follows:

**Experiment 1.** Up–down stair signals were applied to the ESS that responds to the fast acceleration and deceleration processes over different angular control ranges, as shown in Figure 11a.

**Experiment 2.** Trapezium signals with a slope of 30 deg/s were applied to the ESS to verify the response speed of the servo during constant acceleration and deceleration, as shown in Figure 12a.

**Experiment 3.** A chirp signal varying from 1 Hz to 10 Hz was applied to the ESS to verify the dynamics and tracking performance of the servo system during dynamic motion process, as shown in Figure 13a.

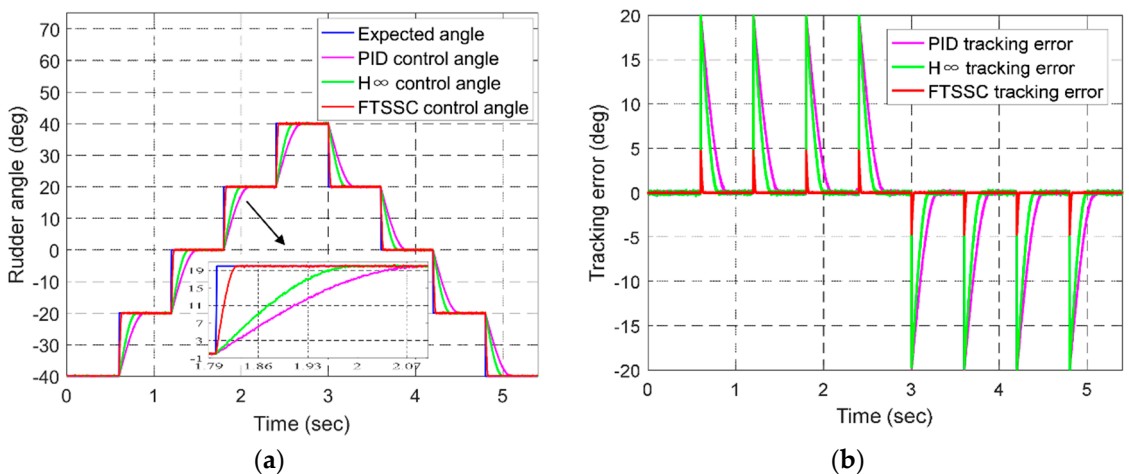

**Figure 11.** Experiment 1 result: (**a**) actual output; (**b**) tracking error.

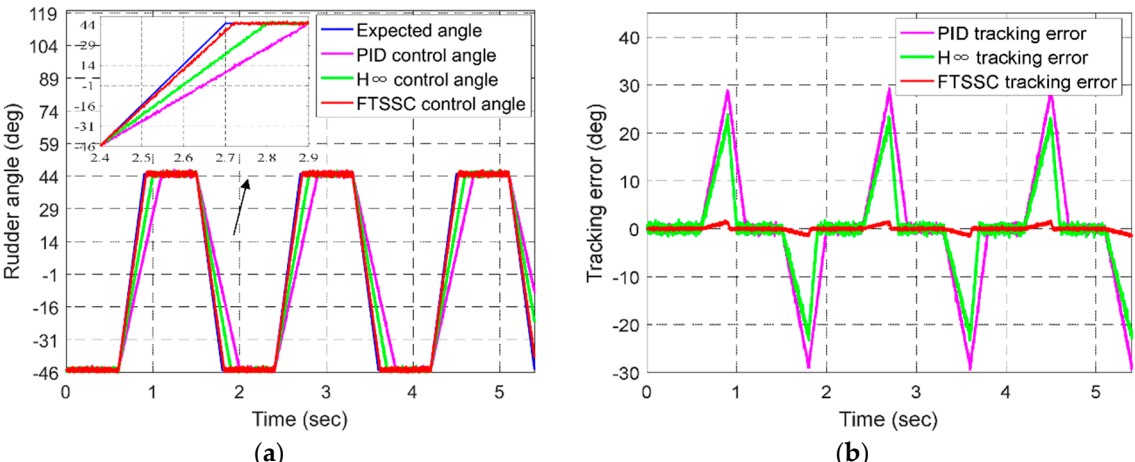

**Figure 12.** Experiment 2 result: (**a**) actual output; (**b**) tracking error.

From Figures 11a, 12a and 13a, it can be seen that compared with the existing PID control and H∞ control law, the FTSSC strategy proposed in this paper can follow the given signal quickly and accurately, whether in the rapid acceleration and deceleration process in different signal ranges, the acceleration and deceleration process changing at a constant angle, or during dynamic motion process.

Figure 11b, Figure 12b, Figure 13b show that the control error of FTSSC was reduced by 75% compared to PID and H∞ control law, which verifies the effectiveness of the proposed strategy.

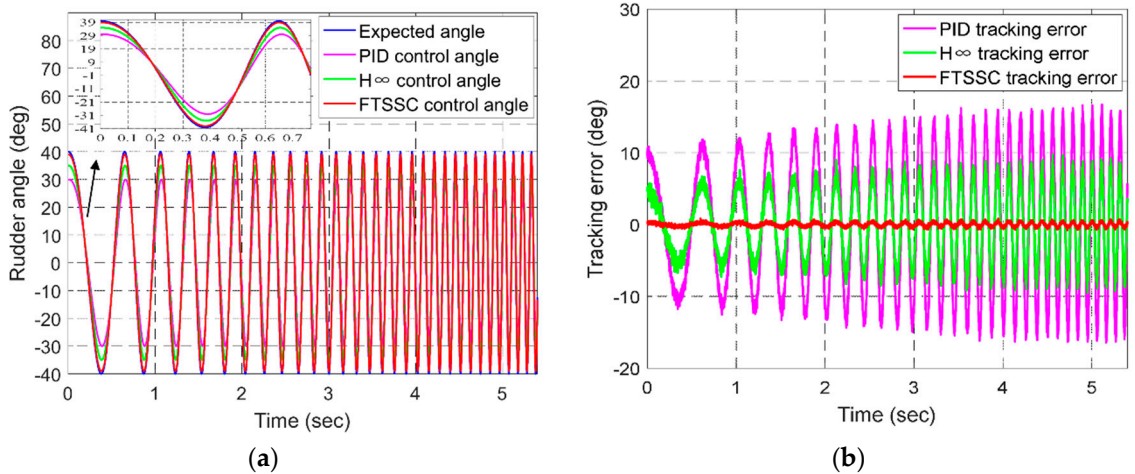

**Figure 13.** Experiment 3 result: (**a**) actual output; (**b**) tracking error.

In addition, in order to verify the advantages of the proposed controller in energy consumption, Figure 14 shows the control voltage consumption of the controller under different conditions in Experiments 1–3. It can be seen that ESS based on FTSSC controller has lower energy consumption than the other two controllers. Therefore, the proposed control strategy is economical and efficient.

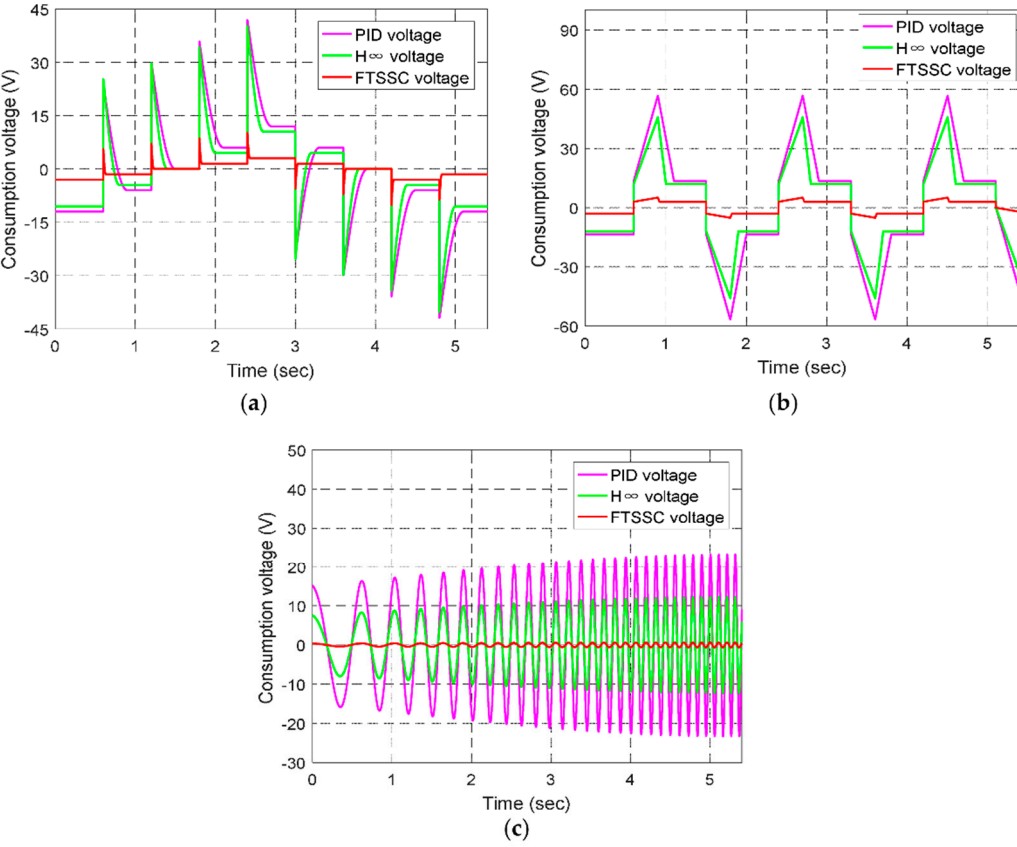

**Figure 14.** Energy consumption results of different control strategies: (**a**) experiment 1; (**b**) experiment 2; (**c**) experiment 3.

Table 3 summarizes the dynamic and static performance response results of Experiment 1–2, including setting time $t_s$, overshoot $\sigma_p\%$, and steady state error $e_{ss}$. As can be seen from Table 3 and the histogram of Figure 15, all steady-state times were within 100 ms, while all overshoots during acceleration and deceleration were less than 3%. Compared with the other two control methods, the control accuracy was improved by at least 50%, which indicates that the proposed control strategy can effectively and accurately achieve rudder angle tracking, thus meeting the requirements of transient and static performance under actual operating conditions.

**Table 3.** Analysis and summary of experimental results.

| Experiment | Reference/° | $t_s$/ms | $\sigma_p\%$ | $e_{ss}$ |
|---|---|---|---|---|
| | −40 to −20 up | 70 | 2% | 0.5 |
| | −20 to 0 up | 65 | 1.7% | 0.4 |
| 1 | 0 to 20 up | 68 | 2.5% | 0.2 |
| | 20 to 40 up | 80 | 1.9% | 0.3 |
| 2 | −45 to 45 ramp | 90 | 1.5% | 0.5 |
| | 45 to −45 ramp | 95 | 1% | 0.6 |

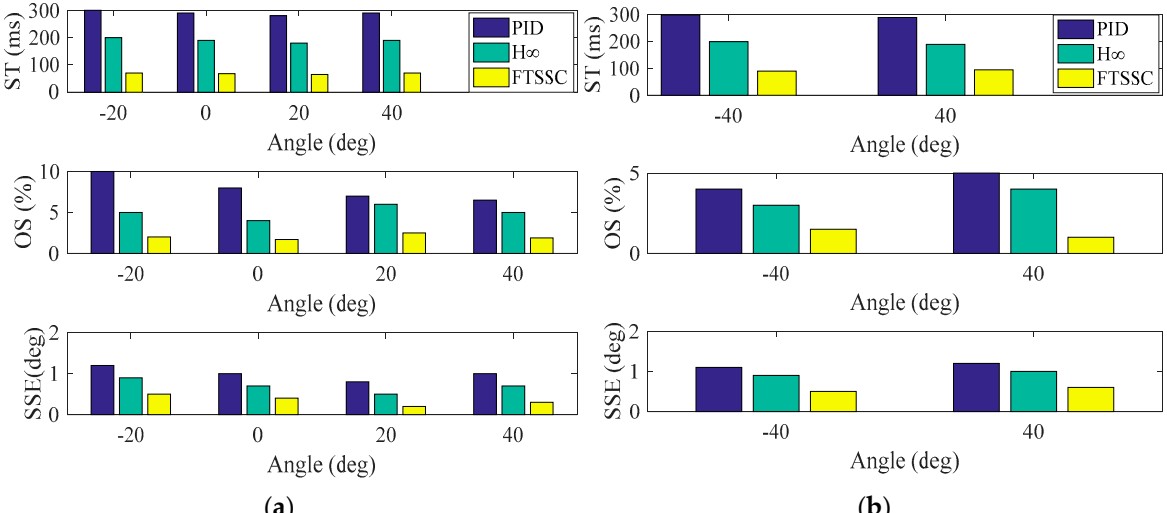

**Figure 15.** Performance comparison of different control strategies: (**a**) experiment 1; (**b**) experiment 2.

## 5. Conclusions

In this paper, the difficult problem of improving ESS resp onse accuracy and response speed was influenced by uncertainties related to friction, clearance, and component aging. In order to overcome these difficulties, the system controller design and the actual physical parameters of the ESS can be accurately and quickly identified. The GA algorithm was introduced into the PSO algorithm and the inertia weight of the PSO was improved to obtain the accurate mathematics of the ESS. A finite-time servo controller for ESS was designed based on the finite-time stability of the model, which converges faster than the asymptotic stability. Co-simulation experiments based on ADAMS and MATLAB/Simulink showed that this strategy can ensure satisfactory tracking accuracy and speed. Meanwhile, the electric steering gear under different conditions was tested on the ESS test platform to verify the effectiveness and practicability of the control strategy. However, in the current work, online identification and tracking of ESS parameters in dynamic environments remains to be further studied.

**Author Contributions:** All authors were involved in the study in this manuscript. Z.W. made an idea and wrote the paper; R.Y. and C.G. provided the theory guidance; S.G. confirmed the theory; X.C. and Z.W. wrote software programs and implemented experiments. Z.W. carried out the experiment.

**Funding:** This work was supported by National International Science and Technology Cooperation, project grant number 2014DFR70650, and the fund for Shanxi "1331 Project" Key Discipline Construction Plan.

**Conflicts of Interest:** The authors declare no conflict of interest.

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
