# Peer review of "Analysis and Verification of Finite Time Servo System Control with PSO Identification for Electric Servo System"

_energies, doi:10.3390/en12183578_

Round 1

Reviewer 1 Report

"1. Introduction " This section contains a very brief overview. For each source to give a detailed answer. Implementation and disadvantages. You can make a comparative analysis.

"2. Preliminaries " section must be renamed

line 119  justify the formulas.

"Figure 4. Flowchart of the improved particle swarm algorithm 232"  Format the algorithm according to international rules. The algorithm is very truncated.

 the scheme needs to clarify and justify the blocks for MATLAB/Simulink 

Reviewer 2 Report

The article presents a adaptive control approach to servo control system. The model parameters are estimated using Particle Swarm Optimization and Genetic Algorithms while the control design for reference tracking is achieved using a finite time servo system control. While the approach is a well balanced with theory, simulations and experiments, I think the article can be improved in significantly. Here are my comments:

In the introduction I do not see any connection of the theme of the manuscript with the theme of the energies journal. Please bring in this perspective otherwise the article would be misfit for this journal and may fit into another journal of MDPI.  The authors use the phrase 'globally finite time stable' in Lemma 1. Please define rigorously what is meant by 'globally finite time stable'.  The control design guarantees global finite time stability, how can we guarantee that once the error is zero in finite time, it will continue to be zero after that? In eqn. (7), the authors should state the dynamical system under consideration in state space form. The states should be θ, ω, θm and ωm. Please state the four equations of the dynamical system in state space form in eqn. (7). The approach followed by authors falls in the paradigm of certainty equivalence adaptive control, where the parameters are estimated and then the estimated parameters are used in the control law. Please make appropriate connections to adaptive control theory in the introduction. In the results section, there is no comparison with existing approaches as discussed in the introduction. Without a comparative study, it is difficult to understand the benefits of closed loop performance of the method presented here over prior work. Overall I think the article needs correction of English grammar at several places. 

Reviewer 3 Report

It will be more interesting if the authors highlight the numerical findings of the work in the abstract section (which shows the improvement of this proposed design).

For Table 3 results, it will be good if the authors can provide some histogram showing the difference of the obtained results side by side using different parameters.

Reviewer 4 Report

General remarks:

The text, especially the first chapters, should be carefully re-written to address the following problems:
a) the optimisation criteria are not well defined,
b) it is not clear if (and how) the method was tested with real controller.

Since the article deals with a optimisation method, it would be desirable to show experimental results before and after controller optimisation. 

The language of the article should be revised.

Some detailed notes (not a complete list):

100...118 - a bit trivial description of a closed-loop system operation; block diagram should be re-drawn to show information flow (also - where is the pivot point of the rudder); 

"desired rudder angle" overused

equation (7) is invalid - remove the sgn(s); this is supposed to be a dynamic model, not a function definition

Lemma 3 - what is a "position real number"

line 153 and 154 - the text states that "PSO is" ... "based on the behavior of PSO"

Figure 3 - The result Xi cannot be derived from performance index - the scheme is incorrect, what are you trying to optimize?

Chapter 3.2.1. Combination with GA: there is nothing on "combination" in the text. The chapter just introduces the genetic algorithm method, with no connection to the main topic. The following graph (figure 4), shows that GA is only used to pull the system out of local minimum - considering the nature of GA, this might not be the most efficient solution. 

Reviewer 5 Report

Review of Analysis and Verification of Finite Time Servo System Control with PSO Identification for Electric Servo System.

The paper is valuable and can be improved in the following aspects:

Using a list of multiple references is not very helpful for a reader. If you need to use more references at least a short assessment/justification should be provided related to the topic of the article. In order to give the readers a sense of continuity, I encourage you to identify publications of similar research in your paper. Please, do a literature check of the papers published on ESS topic in recent years and relate the content of relevant papers to the results and findings presented in your study. A nomenclature should be added, which includes all symbols used and their units. In addition, symbols should be used in a consistent way to differentiate different parameters and avoid confusion. After relation (13) the following expression “where d=1,2,..,D.i=1,2,..m. k represents …… ” should be clarified , maybe a comma is missing after each parameter listing. At page 8 of 15 „For the subsystem (x1 x2) of equation (23), the time derivative of the Lyapunov function 246 is chosen to satisfy the following formula : ” should be reformulated, e.g: „The derivative of the Lyapunov function considered for the nonlinear state space model (23) is chosen to satisfy....”

Round 2

Reviewer 2 Report

I am satisfied with all the responses except the connection of the article with the broader theme of this journal. The authors claim that a better controller can result lower energy consumption and emissions for ESS, but I do not see any study where they try to relate the system performance on energy consumption. If possible, please show the fuel consumption analysis of the various control schemes.

Reviewer 4 Report

The optimization criteria are still not well defined in the updated text with even more confusion introduced by the new text. Do the authors really try to find optimal solution in a 10-dimensional space?

line 54: Authors write: "However, genetic algorithms use cumbersome coding and decoding processes compared to other intelligent algorithms, which may lead to the complexity of the problem."

So why is the genetic algorithm used in this paper?

There is a problem of energy efficiency introduced in the revised paper (in the abstract and line 75), suggesting, that the proposed control is economical. There is no proof of that in the paper (in terms of calculations or measurements).

The language of the article is improved but still needs to be revised.

Figure 2: block diagram contains "Server driver" which is not mentioned in the text, the updated description of the control principle is not better than the original. 

line 210: "When Pg does not change in 5 iterations but does not meet the criteria for optimizing accuracy ..." 
a) why 5 iterations? (Is this valid for all optimization cases, or for a specific one shown oin this paper?)
b) what are the "criteria for optimizing accuracy" and how the authors find them with a real-world object (not a simulation test)?
c) What if the desired accuracy is unreachable?

Chapter 3.2.1. Figure 5 shows a parallel use of PSO and GA, not a "combined" or "hybrid" algorithm. Also "random initialization of parameters" suggests that GA is used to re-initialize the search process with new set of solutions (vector X). 

Formula numbering is broken in the updated text - numbers are incorrect after (42) on page 7.

Is the fitness value defined in (16) also used in Figure 10?

Figure 10 is misleading - since all methods are based on random data (at initialization), the relations between then, can also be random. 

Figures 11, 12: plot for PID error (bright yellow) is almost invisible

Round 3

Reviewer 4 Report

The article has been significantly improved.

However, in my opinion, the proposed solution (the optimization process and the controller) works only in a specific, laboratory case. The idea cannot be used in real world because the optimisation criteria (error) is based on known values (pre-set in the model). The optimization defined like this, cannot work with the unknown parameters of the real mechanism. 
